# Hydrocarbon Lubricating Oils with Admixture of Ionic Liquid as Electrorheological Medium

**DOI:** 10.3390/ma16010330

**Published:** 2022-12-29

**Authors:** Tomasz Jan Kałdoński, Jarosław Juda, Piotr Wychowański, Tadeusz Kałdoński

**Affiliations:** 1Faculty of Mechanical Engineering, Military University of Technology, 00-908 Warsaw, Poland; 2The General Staff of the Polish Armed Forces, 02-519 Warsaw, Poland; 3Oral Surgery and Implantology Unit, Division of Oral Surgery and Implantology, Department of Head and Neck, Institute of Clinical Dentistry, Fondazione Policlinico Universitario A. Gemelli IRCCS, Universita Cattolica del Sacro Coure, 00168 Rome, Italy; 4Departament of Oral Surgery, Medical University of Gdańsk, 7 Dębinki Street, 80-211 Gdańsk, Poland

**Keywords:** hydrocarbon lubricating oils, ionic liquids as additives, electrorheological effect

## Abstract

The article describes the results of experimental studies of electrorheological (ER) properties of lubricating oils containing an admixture of an ionic liquid as the electrically active ingredient. The novelty of these studies consists of the use of selected ionic liquids as additives to hydrocarbon oils in order to obtain quasi-homogenous mixtures with electrorheological properties. So far, such studies have not been carried out. Basic research, which consisted in determining the rheological characteristics in the presence of an external direct electric field, was carried out on a specially designed and built stand, which used a modified Brookfield DV-III Ultra viscometer. The conducted research showed that the produced mixtures generated the ER effect in the presence of a direct electric field with an intensity of up to 0.2 kV·mm^−1^. The tested mixtures showed different electrorheological characteristics. The research was also carried out in the so-called dielectric spectroscopy using the Hewlett Packard HP4192A impedance analyzer. The mechanism of generating and decaying the ER effect was diagnosed by in situ microscopy using the Nikon Eclipse LV100D optical microscope. It was found that the course of the *τ* = f($\dot{\gamma }$) characteristic of a mixture of hydrocarbon oil with a small admixture of an ionic liquid is mainly influenced by the so-called dielectric properties of the electrically active component, or rather their change as a function of the applied BIAS (DC) voltage. At the same time, the obtained results of the research gave grounds to state that the electrorheological characteristics also depend on many physicochemical properties of the mixture components and on the differentiation of their values e.g., from the difference in viscosity of the insulating base oil and the added ionic liquid, and also from the difference in the value of the surface tension of the base oil and the added ionic liquid. In these studies, it was found that the surface tension of the CJ001 ionic liquid at 25 °C was 26.032 mN·m^−1^. The surface tension of CJ008 was 28.099 mN·m^−1^ and that of PAO-6 oil was almost the same, i.e., 27.523 mN·m^−1^. The first mixture (GP1 + CJ001) showed Bigham characteristics and the second (PAO6 + CJ008) Newtonian, in the second mixture, the viscosity difference of the components was two times lower than in the first one (GP1—12.61 mPa·s, CJ001—552.42 mPa·s and PAO6—47.35 mPa·s, CJ008—327.24 mPa·s).

## 1. Introduction

Ionic liquids (ILs) have recently attracted a lot of interest from theorists and practitioners dealing with the rheology of lubricating oils due to their diverse and often unique properties [1,2,3,4,5,6,7,8,9,10,11,12,13,14]. The current definition of ionic liquids states that: “they are all liquid substances consisting only of ions (cations and anions), the melting point of which is lower than 100 °C” [9]. Most commonly used in various applications and scientific research, ionic liquids are formed as a result of the combination of a large organic cation and an inorganic or organic anion. In modern technology, the unlimited possibilities of modifying the structure of the cation and anion provide a large number of potential derivatives with various physicochemical properties. The possibility of combinations is huge and it is estimated at 10^18^ [9,15]. In the near future, it becomes possible to manipulate the properties of ionic liquids produced for a specific purpose. Some successes in the targeted design of ionic liquids (Task Specific Ionic Liquids—TSILs) have been achieved only in relation to solvents [7,9]. The basic and most important physicochemical of ionic liquids, most often mentioned in the literature [4,5,6,7,8,9,10,11,15] and at the same time important from the point of view of the topic of the present article, include: density, viscosity, melting point, vapor pressure, hydrophobicity, and polarity. These properties have a direct and decisive influence on the functioning of all lubricated mechanical systems. Several research centers in the world, including in Japan, China the United States of America, and also in Poland (at the Military University of Technology in Warsaw) have been testing the lubricating ability properties of ionic liquids as separate lubricants or as additives improving the lubricating properties of industrial oils, among others [1,2,3,4,5,6,7,10,12]. In a few works on the use of a specific ionic liquids, the authors, unfortunately, do not specify its type, nor do they describe the method and condition of the mixture obtained (solution or emulsion?). Therefore, it is not known whether the results presented by them relate to a homogeneous mixture (solution) or to a mixture that was a colloidal emulsion, i.e., a quasi-homogeneous liquid. These properties are important for all lubricated mechanical systems, including electrorheological systems. Therefore, from the point of view of the subject of the present article, it is interesting whether a properly selected ionic liquid introduced in a small concentration into a lubricating oil (e.g., base, hydraulic, machine…), can allow for the production of a mixture with electrorheological (ER) properties. However, in any case, the specific miscibility of the ionic liquid with the oil is very important. Some ionic liquids have the ability to dissolve many organic and inorganic materials, but most ionic liquids do not mix with most liquid hydrocarbons [9]. In the present research, the aim was to accurately determine the miscibility of selected (available) ionic liquids with conventional organic and synthetic lubricating oils, and then to assess the electrorheological properties of the mixtures produced. Information on the excellent lubricating ability properties of ionic liquids is provided in the literature, e.g., [1,2,3,4,5,6,7,10,11,12,13]. However, there is no information on the possible use of an ionic liquid as an electrically active component in a mixture with electrorheological properties. Thus, the primary objective of the research described in this article was to check whether a hydrocarbon oil containing a small amount of a selected (well-miscible) ionic liquid, under the influence of an external electric field, can show the ER effect.

## 2. Materials and Methods

### 2.1. Assessment of the Liquid Miscibility

As already mentioned, ionic liquids have a high dissolving capacity for many materials, but at the same time, most of them are not miscible with most liquid hydrocarbons. According to [9], this undermines the generally accepted principle that “like dissolves like”. Apart from extensive tribological tests of ionic liquids, the results of which were presented, among others, in the works [1,2,3,4,16], the miscibility of these liquids in mineral and synthetic oils was also tested [14]. Five lubricating oils (PFPE Y04, GP1, PAO6, SN-65,0 and Hipol 15F SAE 85W/90) and seven ionic liquids marked with symbols CJ001-CJ008 were subjected to detailed tests of miscibility.

PFPE Y04 oil is a synthetic perfluoropolyether oil by Solvay Solexis [17,18]. GP1 oil is a silicone hydraulic damping fluid that is a mixture of polymethylsiloxanes and marked in the Polish Army with the code MPS S-91715 [19]. PAO6 is a polyalphaolefin base oil [20]. SN-650 is a mineral base oil from the vacuum distillation of crude oil [21]. Hipol 15F SAE 85W/90 is a gear oil used for the lubrication of automotive vehicle transmission [22]. The ionic liquid CJ001 is 1-methyl-3-octyloxymethylimidazole tetrafluoroborate [23]. CJ002 is 1-methyl-3-octyloxyimidazole bis(trifluoromethylsulfonyl)imide [23]. CJ003 is 1-methyl-3-butoxymethylimidazole bis(trifluoromethylsulfonyl)imide [23]. CJ004 is a 3-butoxymethyl-1-butylimidazole bis(trifluoromethylsulfonyl)imide [23]. CJ006 is 3-methyl-1-propylpyridine bis(trifluoromethylsulfonyl)imide [15]. CJ007 is 1,2-dimethyl-3-propylimidazole bis(trifluoromethylsulfonyl)imide and CJ008 trihexyltetradecylphospho nium bis(trifluoromethylsulfonyl)imide [15]. The previously mentioned studies [1,2,3,4,16], which confirmed the excellent tribological properties of ionic liquids, were the basis for the above selection.

Initial miscibility assessment was performed under a microscope observing the “bonding” of the ionic liquid to the oil at the bottom of the watch glass. In cases of effective mixing of the ionic liquid with the oil, 50 mL samples were made in glass flasks, with 5% (*v*/*v*) content of the selected ionic liquid. Subsequently, the liquids were mixed for 20 min with a mechanical agitator and sonicated for another 20 min in VTU SCB ultrasonic washer at 50 °C. After mixing the samples were allowed to stand for 48 h, observing whether there were any precipitation or sedimentation. In the absence of such symptoms, the mixture was qualified for further testing.

### 2.2. Identification of the Basic Physicochemical Properties of the Liquid

The basic physicochemical properties of mixtures (and their components) qualified for electrorheological tests were assessed. The assessment of density (*ρ*), kinematic viscosity (*υ*) and dynamic (*μ*), viscosity index (*VI*), surface tension (*σ*), refractive index (*n_D_*), and specific refraction (*r_D_*) of the liquid was carried out. It is important to know the variability of these properties in the case of oils containing a certain amount of ionic liquid with significantly different properties. Immediately before the measurement, the samples of mixtures of oils with ionic liquids were again sonicated for 20 min at room temperature (about 22 °C).

The KSV Sigma 701 (Espoo, Finland) tensiometer was used to measure the density (*ρ*) and surface tension (*σ*). The entire course of measurements was controlled by computer software. Measurements were made according to the instructions of this device [24] at the temperature of 25 °C, 40 °C, and 100 °C. The samples were thermostated using a Julabo F12 bath [25].

The kinematic viscosity (*υ*) was measured in accordance with PN-EN ISO 3104:2004 [26], and the viscosity index (*VI*) was calculated in accordance with PN-EN ISO 2909:2009 [27]. To measure the kinematic viscosity of oils, the popular TAMSON TV 2000 thermosta-tic bath with a Pinkievich viscometer [28] was used, which was carried out at the temperature of 25 °C, 40 °C, and 100 °C. Due to the limited amount of very expensive ionic liquids, their viscosity was measured with the AMVn micro viscometer by Anton Paar (Graz, Austria) [29].

Abbe’ laboratory refractometr RL1-PZO (Poland) was used to determine the refractive index (*n_D_*) [30]. The specific refraction (*r_D_*) was calculated using the commonly known formula:(1)$$r_{D}=\frac{n_{D}^{2}-1}{n_{D}^{2}+2}\cdot \frac{1}{\rho }$$

### 2.3. Evaluation of Electrorheological Properties of Liquid Mixtures

For such an assessment, a viscometer is necessary, adapted to register changes in the shear stresses (viscosity) of the liquid under the influence of an external electric field. Due to the lack of access to such a specialized device, an own electrorheological test stand was designed, built, and launched on the basis of the available standard Brookfield DV-III Ultrarotational viscometer, which made it possible to meet the basic requirements for conducting this type of research, including:(a)liquid flow parameters in the rheometer measuring the gap and measurement conditions should be selected so that the flow is steady, laminar, and isometric;(b)the design of the rheometer should ensure a homogeneous electric field throughout the gap;(c)the rheometer must enable measurement at very low shear rates where the ER effect is most visible;(d)it should be possible to control the operation of the viscometer by means of a PC with specialized instrumentation.

Details of modification and adaption to ER testing of the Brookfield DV-III Ultra rotational viscometer are described in the works [14,31]. Figure 1 shows the stand used to test the electrorheological properties of prepared mixtures of oil with ionic liquids.

Specialized software Reocalc 32 v.2.6 made it possible to work in the automated measurement mode, which is recommended in this type of research. Thanks to this, the measurements, could be carried out with the automatic setting of selected angular velocities of the spindle rotation, with the activation of the measured data, and with the recording of the results in disk sets. On the prepared stand (Figure 1), commissioning and control tests were performed, the purpose of which was to confirm the correctness and accuracy of the measuring system operation. For these tests, the commercial two-phase electrorheologicalliquid LID 3354S from the English company SMART TECHNOLOGY Ltd. (Solihull, UK) was used. This ER liquid consists of silicone oil and polymer particles with a diameter of 45 μm in the amount of 37.5% (m/m) [32]. The result of these studies was very positive–in the presence of a direct electric field, the Bingham characteristics (*τ = τ_o_* + *η*·$\dot{\gamma }$) was obtained, typical for such a heterogeneous ER liquid. Therefore the research kit presented in Figure 1 was submitted to the Patent Office of the Republic of Poland, which on 10 February 2017 issued a decision on granting a Protective Law.

### 2.4. Studies of Liquids in Dielectric Spectroscopy

Such tests allow us to assess the variability of the dielectric constants (*ε*) as a function of the frequency (*f*) of the BIAS (DC) electric field for different values of the applied voltage (*U*), as well as the simultaneous variability of resistances (*R*) and conductivity (*κ*). It was also planned to repeat the tests with the same samples to observe how the originally applied BIAS (DC) field affects the further behavior of the liquid in the electric field. These tests were performed using the HP4192 impedance analyzer by Hewlett Packard (Palo Alto, CA, USA) [33], in the frequency range 100 Hz~10 MHz. The applied BIAS (DC) field with a different value for the voltage *U* = 0~35 V made it possible to determine the influence of an external factor on the tested sample. The alternating measuring field (AC) was defined with a voltage of 0.1 V. Measurements were made at a constant room temperature (about 22 °C). Cells ITO with electrodes made of In_2_O_3_ doped with 10% (m/m) SNO_2_, with the resistance of the order of 100 Ω, were used to perform the measurements. The *f_o_* cut-off frequency for ITO cells was about 400 kHz, which is the frequency at which the capacitor’s ability to charge and discharge is reduced. Below this frequency, the capacitor cannot fully charge. Cells of different thicknesses (about 5, 12, and 50 μm) were used in the research due to the differrent composition and types of the tested mixtures and their components, e.g., for the LID 3354S standard liquid containing polymer particles, the thickest cells were used. For particulate free liquids, thinner cells were used. These cells consisted of two plane–parallel glass plates. The electrodes were sprayed onto the glass and received the signal from the analyzer. So-called glass spacers were used to obtain different thicknesses of the measuring cells. Due to the specificity of the measuring cells, the USS 9200 ultrasonic soldering iron of the Swiss company MBR Electronics was used for soldering the constants. The cables with a diameter of 0.25 mm, carrying the measuring signal of different frequencies, were made of silver-plated oxygen-free copper.

### 2.5. Microscopic Study of the Mechanism of the ER Effect

In order to recognize the mechanism of the ER effect, its formation, and course, in situ microscopic examinations of the prepared mixtures were carried out under the action of an external direct electric field. The microscopic observations were performed on the Japanese optical microscope Nikon Eclipse LV100D (Tokyo, Japan) [34]. For recording photos and video, themicroscope was coupled to a PC with NIS-AR image analysis software installed. For in situ observations, a “vessel” with electrodes was made (Figure 2) and filled with the tested liquid. The electrodes were supplied with direct current by the HCP 14-6500 high voltage power supply from FUG Elektronik GmbH (Schechen, Germany), which was previously a component of the stand with the modernized Brookfield DV-III Ultra viscometer (Figure 1). The observation was carried out in transmitted light the lens magnification ×50, using the bright field technique, at a temperature of about 25 °C and air humidity in the laboratory of about 55%.

## 3. Results

### 3.1. Results of the Liquid Miscibility Assessment

As a result of miscibility tests of selected hydrocarbon oils and ionic liquids, it was found that most of the mixing attempts, were ineffective. Only two homogeneous mixtures were obtained, the first mixture was GP1 hydraulic oil with CJ001 ionic liquid and the second was PAO6 base oil with CJ008 ionic liquid. In both cases after mixing, according to the methodology described in Section 2.1, oil with 5% (*v*/*v*) content of ionic liquid, a very good combination was achieved. After waiting for 48 h, no color change, no precipitation or sedimentation of the ionic liquid was noticed. These two mixtures and their component were subjected to further tests.

### 3.2. Results of Identification of the Basis Physicochemical Properties of the Liquids

Table 1 presents the results of tests of the basic properties of mixtures and their ingredients, at a temperature of 25 °C, important in the analysis of electrorheological properties of lubricating liquids [1,14,16,35]. Figure 3 shows the variability of the liquid density and their viscosity in temperature-rise function, typical for Newtonian liquids.

### 3.3. Results of the Evaluation of Electrorheological Properties of Liquid Mixtures

Studies of electrorheological properties of GP1 + CJ001 and PAO6 + CJ008 mixtures were preceded by preliminary tests to select a suitable “viscosimeter spindle/cylinder with electrodes” set (Figure 1) enabling optimal testing. Table 2 shows possible uses in modernized viscometer sets “spindle/cylinder” and their designation [14,36].

To realize the full test cycle of electrorheological properties of both mixtures, JJ6 and JJ8 sets were selected, which allowed the relatively smallest shear speeds $\dot{\gamma }$ [s^−1^]. During preliminary tests, it turned out that the produced mixtures made of 5% (*v*/*v*) of the content of ionic liquid have too much electrical conductivity, which led to short circuits. Therefore, to test ER effect, the content of ionic liquid to 2% (*v*/*v*), obtaining a positive effect. The results obtained in the Reocalc 32 v.2.6 program have been collected in Table 3 and Table 4 and presented graphically in the collective drawing (Figure 4).

### 3.4. Results of Liquid Studies in Dielectric Spectroscopy

Figure 5, Figure 6, Figure 7, Figure 8, Figure 9, Figure 10, Figure 11 and Figure 12 show selected characteristics *ε* = f(*f*) and *κ* = f(*f*), at different BIAS (DC) voltage values. Usually, the literature gives the values of, *R* and read for the frequency *f* = 1 kHz. They are treated as so-called “real values”, facilitating the initial analysis of the state defined by these values. Such preliminary analysis may be adequate only for pure dielectric and insulating liquids with a very small and stable values of *ε*, *κ* and *R*. In the case of the tested ionic liquids and their mixtures with hydrocarbon oils, a detailed analysis of the course of these characteristics as a function of frequency (*f*) and voltage (*U*) of the applied direct current (DC) must be carried out.

In order to more accurately recognize the influence of lower BIAS voltage values on the course of the characteristics *ε* = f(*f*), the ionic liquid CJ008 was additionally tested in dielectric spectroscopy with ITO cells with a thickness of 12 μm. The measurement was performed in the range of the same frequency as before, i.e., 0.1~10,000 kHz, but for lower BIAS voltages, i.e., from 0 to 10 V (0, 1, 2, 5, 7, and 10 V). For these tests, the ionic liquid CJ008 was used, with a tendency to change (*ε*) less than CJ001 under the influence of BIAS voltage (Figure 5a and Figure 6a) and giving in a mixture with PAO6 three times greater ER effect than the mixture of GP1 oil and CJ001 ionic liquid (Figure 4). A selected example of measurements of the real part of the electric permeability (*ε*) of the CJ008 ionic liquid as a function of the frequency (*f*) of the measurement field for small BIAS (DC) voltages in the range *U =* 0~10 V is presented below (Figure 12a). In order to observe whether the applied BIAS field affects the “further history” of the sample, a second (subsequent) measurement was performed for the same sample of the CJ008 ionic liquid–for the same BIAS fields (Figure 12b).

### 3.5. The Results of Microscopic Studies of the Mechanism of the ER Effect

First, in situ microscopic observations of commercial liquid LID 3354S, which acted as a standard and reference liquid, were performed. The following photos (Figure 13a–f) show the characteristics sequences showing changes in the internal structure of the LID 3354S liquid as the intensity of the constant electric field increases.

The procedure of testing mixtures containing ionic liquids was similar. The following photos (Figure 14a–j) show selected characteristics sequences showing changes in the internal structure of the mixture of the base PAO6 polyalphaolefin oil as a matrix and 2% (*v*/*v*) of the CJ008 ionic liquid, i.e., trihexyltetradecylphosphonium bis (trifluoromethylsul-fonyl) imide, treated with external electric field with the intensity *E =* 0~0.3 kV·mm^−1^.

The next photographs (Figure 15a–h) show selected characteristic sequences showing changes in the internal structure of the mixture of the silicone damping liquid GP1 and 2% (*v*/*v*) of the CJ001 ionic liquid, i.e., 1-methyl-3-octylmethylimidazole tetrafluoroborate, treated with external electric field with the intensity *E =* 0 ~ 0.3 kV mm^−1^.

## 4. Discussion

The results of the conducted research suggest that obtaining a homogeneous mixture with ER properties, made of hydrocarbon oil as an insulating liquid and ionic liquid as an electrically active additive, is complicated. First, it is very difficult to select an ionic liquid that is fully miscible with a hydrocarbon liquid, which is a good electrical insulator. There is no major problem with selecting a hydrocarbon liquid with excellent insulating properties (e.g., PAO6 and GP1 oils selected in these tests). It is more difficult to choose an ionic liquid, a small amount of which will allow the production of a homogeneous mixture (solution or colloidal emulsion of liquid components) with ER properties. Such a liquid should have, among others: low viscosity in the condition of no action of the electric field, a large increase in shear stress as a result of the action of the direct electric field, short reaction time to the action of the electric field, the ability to work in a wide temperature range. Inhomogeneous liquids, there are no unfavorable phenomena typical for heterogeneous liquids, such as coagulation and sedimentation, related to the presence of solid particles in the liquid. However, a significant limitation in the practical application of homogeneous ER liquids is their high sensitivity to contamination and tendency to electrical breakdown (short circuit). Two mixtures selected for ER research, i.e., GP1 + CJ001 and PAO6 + Cj008, met almost all the requirements (Section 3.2), but it turned out that both ionic liquids had too high electrical conductivity (Section 3.4—Figure 5a,b) which, with their higher content in the base oil, was the cause of breakdown during the action of an electric field. However, reducing the content of the ionic liquid in the oil to 2% (*v*/*v*) made it possible to obtain mixtures with ER properties, but the obtained ER effect was short-lived and unique for the same liquid sample, which suggested irreversible changes in the internal structure of these mixtures under the influence of an external electric field. A better effect of ER (higher values of *τ* and *η* during the action of the external electric field-Table 3 and Table 4) was obtained for the PAO6 + CJ008 mixture.

The tested mixtures showed different rheological characteristics of *τ* = f($\dot{\gamma }$), both in the absence of an external electric field and in the presence of this field (Figure 4). The GP1 + CJ001 mixture, in the absence of an electric field, behaved similar to Bingham’s liquid (*τ = η*·($\dot{\gamma }$) + *τ_o_*), but its limit stress (*τ_o_*) had a low value (approx. 0.5 Pa). In the presence of an electric field, up to the value of *E =* 0.2 kV·mm^−1^, the rheological characteristics of this liquid could be described by Herschel-Bulkley equation *τ = m_1_*·($\dot{\gamma }$)^n1^ + *τ_o_* with *n* < 1 typical for a viscoplastic pseudoplastic liquid. The second mixture, i.e., PAO6 + CJ008 at *E =* 0 V behaved similar to a Newtonian liquid *τ = m_2_*·($\dot{\gamma }$)^n2^ with *n* < 1 typical for viscous pseudoplastic liquid (Figure 4). After increasing the electric field strength to a value of *E =* 0.3 kV·mm^−1^, the rheological characteristics of both mixtures corresponded to those of the mixtures without the influence of the electric field, i.e., the mixture GP1 + CJ001 to the Bingham characteristics and the mixture PAO6 + CJ008 to the Newton characteristics. It should be remembered at this point that the measurements of the ER properties of these mixtures with the modernized Brookfield DV-III Ultra viscometer were carried out at the lowest possible shear rate for the GP1 + CJ001 mixture it was ($\dot{\gamma }$)_min_ = 4.20 s^−1^ (Table 3), and for the PAO6 + CJ008 mixture it was ($\dot{\gamma }$)_min_ = 2.67 (Table 4). Obtaining even lower shear rates, with the determined selection of the “spindle/cylinder” sets, was impossible. Therefore, the real course of the rheological characteristics *τ* = f($\dot{\gamma }$) for even lower shear rates is practically impossible to determine, however *τ* = f($\dot{\gamma }$) courses established for both mixtures are the most probable (Figure 4).In both cases, at *E =* 0.3 kV·mm^−1^, the viscosity (*η*) again corresponded to the value at *E =* 0. The maximum values of viscosity, caused by the action of the electric field, were obtained in both cases at the lowest shear rates, when the electric field intensity was *E* = 0.2 kV·mm^−1^. In the case of the mixture GP1 + CJ001 it was the value of *η* = 435.71 mPa·s, which was almost 2.5 times higher than the viscosity in the absence of an electric field (Table 3). However, in the case of the PAO6 + CJ008 mixture, the components of which are more polar than the GP1 + CJ001 mixture (Section 3.2—Table 1), the effect of action of the electric field was greater. Viscosity increased to the value of *η* = 1247.19 mPa·s (Section 3.3—Table 4), which was almost 20 times higher than the value of (*η*) in the absence of electric field. Thus, a much better ER effect was obtained for the PAO6 + CJ008 mixture described by the Ostwald-de Waele equation *τ = m*_2_·($\dot{\gamma }$)^n2^, the components of which different in viscosity value by 279,88 mPa s, i.e., two times less than in the GP1 + CJ001 mixture described by the Herschel-Bulkley equation *τ = m*_1_·($\dot{\gamma }$)^n1^ + *τ_o_*, where the difference was 539,81 mPa·s. It should also be noted that in the first case the base insulating liquid, i.e., PAO6, had a viscosity *η* = 47.36 mPa·s, and in the second case the base insulating liquid, i.e., GP-1, had a viscosity *η* = 12.61 mPa·s, i.e., four times lower than PAO6. Such observation suggests that in discussed case, the effect on the maximum three times the greater value of the viscosity of the PAO6 + CJ008 mixture (*η* = 1247.19 mPa·s), with *E* = 0.2 kV·mm^−1^ (Table 4), compared to the GP1 + CJ001 mixture (*η* = 435.71 mPa·s), with the same value of *E =* 0.2 kV·m^−1^ (Table 3), had among others, the higher viscosity of the PAO-6 base liquid than GP1, and not different viscosity of both components of the mixture, which in turn could be associated with the occurrence of the boundary stress (*τ_o_*) in the GP1 + CJ001 mixture (Figure 4).

Ionic liquids qualified for ER research belonged to the set of liquids previously tested as perspective lubricating oils [1,2,3,4,16]. In those studies, excellent tribological properties of many ionic liquids were found, therefore an attempt was made to use them also as an electrically active additive in a mixture with an insulating hydrocarbon lubricating oil. In order to identify the causes of the formation and disappearance of the ER effect in such a mixture, it was decided to conduct additional tests in dielectric spectroscopy of the components of both discussed mixtures, i.e., base oils GP-1 and PAO-6 and ionic liquids CJ001 and CJ008 (Figure 5, Figure 6, Figure 7, Figure 8, Figure 9, Figure 10, Figure 11 and Figure 12). The obtained results determining the dielectric constant (ε) show that the ionic liquid CJ008, i.e., triheksyltetradecylphosphonium bis(trifluoromethylsulfonyl)imide, showed the highest value, which at *f* = 1 kHz and BIAS (CD) voltage zero was *ε* = 1127, while the ionic liquid CJ001, i.e., 1-methyl-3-oktyloxymethylimidazole tetra-fluoroborate, had this value slightly lower, i.e., *ε* = 1035 (Figure 5a and Figure 6a). Increasing the BIAS (DC) voltage caused a decrease in the value of (*ε*), much more intense for the CJ001 Ionic liquid (Figure 5a) than the CJ008 ionic liquid (Figure 6a). This differentiation may result from the different number of paired ions in both liquids [9]. Such ions have relatively low mobility and contribute to the permeability and relatively low frequencies–at higher frequencies, this effect disappears–faster in the CJ001 ionic liquid than in the CJ008 ionic liquid. This may be another indirect reason for obtaining a weaker ER effect for the mixture containing the ionic liquid CJ001. The values (*ε*) of this liquid, at *f* = 1 kHz and for *U*
≥ 10 V, reached the range typical for dielectric and insulating liquids (Figure 5a). In the case of CJ008, only the voltage *U* ≥ 16 V caused a significant decrease in (*ε*)—Figure 6a. The characteristics of *ε* = f (*f*) and *κ* = f (*f*) for ionic liquids show that measurements above 10 kHz have dubious analytical usefulness, and above 100 kHz do not make much sense because they are distorted by the specific properties of the ITO cell determined by the so-called cut-off frequency (Section 2.4). The course of the characteristics *ε* = f(*f*)—(Figure 5a and Figure 6a) and *κ* = f (*f*)—(Figure 5b and Figure 6b) proves the strong influence of the BIAS (DC) field on ionic liquids and confirms the earlier assumptions that the large BIAS (DC) field was had a destructive effect on the tested ionic liquids and changed their properties. Figure 12a,b show how the applied BIAS (DC) field influenced the further history of the sample of the CJ008 ionic liquid, i.e., the one that gave three times greater ER effect in the mixture with PAO6 than in the mixture with GP1 oil. Measurements as shown in Figure 12a,b were repeated many times, obtaining results showing the same tendency. Treatment of the CJ008 ionic liquid with the BIAS field of low voltage (1~2 V) did not cause large changes, therefore the electric permittivity characteristics as a function of frequency were then usually similar and sometimes the same. It happened, however, that the characteristics for small voltages in *U* ≤ 5 V were “conquered” (i.e., Figure 5a and Figure 6a)—it could have been caused by the variable mobility of ions. In general, the increase in BIAS (DC) voltage caused the electrical permeability of the liquid was decreased, and this could have a reduction of diversified value course even for the same liquid; for the same reasons above. In Figure 12b, we can see that the earlier (previous—Figure 12a) application of the constant voltage caused large changes. The measured values of the real part of the electric permittivity are many times smaller—in the example given in Figure 12a,b, even almost a hundred times lower for *U* = 0 and *f* = 1 kHz. For this frequency and *U* = 2 V, the value (ε) has reached the range typical for dielectric and insulating liquids. Thus, reapplication of the BIAS (DC) field voltage to the same sample of the CJ008 liquid. Accelerated the process of its destruction and led to the same state as in CJ001, which was so damaged even after a single BIAS field operation.

The PAO6 and GP1 base oils behaved completely differently. Their dielectric constants were stable both as a function of the BIAS (DC) field voltage increase and as a function of its frequency (*f*), when a wide plateau was recorded, even above 100 kHz (Figure 7a and Figure 8a), and amounted to 2.02 and 2.30 respectively. This means that the dielectric properties of both base oils do not depend on the applied BIAS (DC) field. Increasing the (*U*) value of this field did not cause any changes in the dielectric constants of both base oils.

PAO6 and GP1 are non-conductive, which confirms their suitability as insulating liquids in ER mixtures. In the case of the tested mixtures GP1 + CJ001 and PAO6 + CJ008, a slight increase in the value of the dielectric constant (*ε*) was observed in the area of a stable plateau (Figure 9a and Figure 10a). For the mixture GP1 + CJ001, this increase was 0.17 in relation to the base oil GP1 for which *ε* = 2.30. A slightly greater increase in the dielectric constant value was recorded for the PAO6 + CJ008 mixture, i.e., by 0.26 in relation to PAO6 base oil, for which *ε* = 2.02. In this case, the plateau was not so stable anymore. There was a slight decrease (*ε*) in the range of 0~100 kHz and at the same time unstabilized values (*ε*) for different values (*U*), not necessarily related to the voltage increase. From a dielectric point of view, this slight admixture of ionic liquids practically did not cause any significant change. Increasing the BIAS (DC) voltage value from 0 to 20 V did not significantly affect the results of the measurements of the dielectric constant of the tested mixtures (Figure 9a and Figure 10a), as well as pure base oils (Figure 7a and Figure 8a). However, in the case of the PAO6 + CJ008 mixture (Figure 10a), it can be seen that under the influence of the voltage *U* > 0, alternately decreases and increases in the value (ε) were noted, denoting a continuous change in dielectric properties caused by probably chaotic movement of ions.

Figure 11a shows, for comparative purposes, the dependence (*ε*) on frequency (*f*) for ER LID 3354S liquid in the range from 0 to 35 V BIAS (DC) field voltage. This liquid showed a very high resistance to the BIAS (DC) voltage, but it did not plateau as a function of the frequency increase (*f*); moreover, its dielectric constant (ε) recorded at *f* = 1 kHz was a bit unstable (3.15–3.20) when changing the BIAS (DC) field voltage—however, without a clearly marked trend. This symptom was probably the result of the uneven dispersion of the solid polymer particles in the base oil of the LID 3354S liquid. At slightly higher frequencies (*f*), regardless of the applied voltage, it had the same value, steadily decreasing gently in the function (*f*), down to the value of about 2.5 at about 100 kHz (Figure 11a). Simultaneously with the decrease in (*ε*) as a function of frequency (*f*), the conductivity (*κ*) gently increased (Figure 11b) and the resistance (*R*) decreased accordingly. Due to the volume of this article, the resistance values are not shown. The greatest resistance was shown by GP1 silicone base oil, i.e., *R* = 3.2·10^9^ Ω at *f* = 1 kHz and *U* = 0 V. The second liquid in terms of resistance value was the mixture GP1 + CJ001, with *R* = 1.71·10^9^ Ω. The third was the base oil PAO6 with a result of *R* = 9.8·10^8^ Ω and the fourth was the mixture of PAO6 + CJ008 with a result of *R* = 1.14·10^8^ Ω. In both mixtures, the admixture of the ionic liquid caused a decrease in their resistance.

The parameter opposite to resistance (*R*) is conductivity (*κ*). The ionic liquid CJ008 showed the highest value of electrical conductivity: at *f* = 1 kHz and *U* = 0 V was *κ* = 2.73 × 10^−5^ S·m^−1^. In the frequency range *f* = 1~10 kHz (Figure 6b), numerous “defects” of the curves *κ* = f (*f*) were recorded as a result of chaotic movements of ionic in the electric field, more intense with increasing value (*U*). Initially, at *U* = 0 V, the conductivity (*κ*) increased the increase of (*f*). Then, with *U* > 0, the characteristics, *κ* = f (*f*) was “boosted” (Figure 6b). This was especially true of the CJ008 ionic liquid and the PAO6 + CJ008 mixture (Figure 10 b). The second, highest conductivity value was found in the ionic liquid CJ001, for which *κ* = 1.21·10^−5^ S·m^−1^. In Figure 5b we can see that initially at *U* = 0 V, the conductivity (*κ*) increases with increasing frequency (*f*). Then, at *U* = 5 V, there is a “boost” of these values. The subsequent increase of the BIAS (DC) field voltage causes a significant decrease in the conductivity value and at *U* = 20 V and *f* = 1 kHz it was *κ* = 2.18·10^−7^ S·m^−1^ (Figure 10b).

The *κ* = f(*f*) characteristics of pure GP1 and PAO6 base oils, presented in Figure 7b and Figure 8b are typical for dielectric liquids and compatible with *ε* = f(*f*) and *R* = f(*f*) characteristics. The results of in situ microscopic tests of both mixtures, i.e., PAO6 + CJ008 and GP1 + CJ001, subjected to an external electric field, allowed us to recognize the course of the ER mechanism—confirmation of its occurrence and registration of symptoms of changes in the internal structure of the tested mixtures, leading to the disappearance of the ER effect (Section 3.5). The same tests were performed for comparison with the commercial liquid LID 3354S [32]. Selected photo sequences showing changes in the internal structure of the LID 3354S liquid are shown in Figure 13a–f. These photos show the successive phases of the formation of the so-called “fibril chains” from polymer particles present in the silicone oil constituting the matrix (base) of the LID 3354S liquid. In the initial phase of observation (Figure 13a), the electric field intensity was *E* = 0 kV·mm^−1^. Polymer solids were seen quite freely “suspended” in the base liquid. Small and sparse clusters of these particles were caused by their mutual molecular interaction resulting from their very high content (close to each other) in the base oil (37.5% *v*/*v* [32]). Then, with an increased intensity of the electric field to the value of *E* = 0.1 kV·mm^−1^ (Figure 13b), more and more numerous “chains” forming the structure of the network were observed. While increasing the electric field intensity to the value of *E* = 0.2 kV·mm^−1^ (Figure 13c), the structure of the network became denser. The chains of polymer particles grew thicker and the network denser. While increasing the electric field intensity to the value of *E* = 0.3 kV·mm^−1^ and then *E* = 0.4 kV·mm^−1^ and *E* = 0.5 kV·mm^−1^, the formed network became denser and denser, especially in the zones adjacent to the electrodes (Figure 13e,f). During these tests, the electric field intensity was increased to the value of *E* = 0.6 kV·mm^−1^, but the structure of the network was stable and did not undergo any further changes. This undoubtedly means that all polymer particles present in the liquid have been effectively “used” to build the chains and structure of networks responsible for the ER effect. After de-energizing, the formed network and chains separated and single nano- and microparticles “freely suspended” were visible again. In subsequent tests with the same portion of liquid, the electric field intensity was increased again in the range 0~0.4 kV·mm^−1^. Similar effects were observed in the first trial. Several attempts were made to increase and decrease the electric field intensity (also with changed polarity) for the same sample In each trial, the effects of creating and compacting the formed network of polymer particles in the LID 3354S liquid was predictable and the pattern of their formation was reproducible. The results of in situ tests of the PAO6 + CJ008 mixture (Figure 14a–j), subjected to an external electric field with the intensity *E =* 0~0.3 kV·mm^−1^, were different. This time, a higher electric field intensity was unnecessary because the ER effect disappeared at *E* = 0.3 kV·mm^−1^. As both components of the PAO6 + CJ008 mixture are transparent the observation of changes taking place in it was very difficult, compared to the observation with the reference liquid LID3354S Therefore, and also due to the high dynamics of changes in a short time, the registration of characteristics sequences was very complicated and tedious. However, they managed to capture the moment of formation of “fibril chains” from the CJ008 ionic liquid, somewhat differently than in the heterogeneous liquid LID3354S. Initially, when the electric field strength was zero (Figure 14a), a large dispersion of the CJ008 ionic liquid nanoparticles in the PAO6 base oil was observed, although occasionally larger particles of the ionic appeared, indicating a certain heterogeneity of dispersion (Figure 14b). Then, with an increased electric field intensity to the value of *E* = 0.1 kV·mm^−1^, the first symptoms of the formation of “fibril chains” were recorded, which were created by these slightly larger particles (most visible in the foreground). Figure 14c shows the effect of an electric field of *E* = 0.1 kV·mm^−1^ on single particles of the CJ008 ionic liquid, which elongate and connect in a chain in direction of the field’s action, between the electrodes. The next photo (Figure 14d) shows the formation of new “fibril chains” with the same value of *E* = 0.1 kV·mm^−1^. After a successive increase in the electric field strength to *E* = 0.2 kV·mm^−1^ (Figure 14e, f), the number of “fibril chains” increases, and subsequent particles of the ionic liquid combine into chains between the electrodes. However, there are no symptoms of the formation of a dense “fibrillary network”—as was the case with the liquid LID 3354S. On the other hand, the first single symptoms of ionic liquid accumulation at the electrode were observed (Figure 14f,g). Figure 14h shows the structure of the same chain in its central part. The dark spots visible in the photos, especially at the points of contact of individual particles of the ionic liquid in the “fibril chain”, are probably unidentified nanoparticles of impurities that easily cluster in the area of the strongest molecular and electrical interactions. By analyzing all the photos showing the so-called “fibril chains” of the CJ008 ionic liquid can be said to be rather single chains (unbranched, not linked to other chains) of particles serially connected to each other and “spread” between the electrodes. The particle sizes of the ionic liquid (micro/nano) are varied and random–resulting from the obtained dispersion of individual samples of the CJ008 ionic liquid in a mixture with PAO6 base oil. Further increase of the electric field intensity to the value of *E* = *0.3* kV·mm^−1^ resulted in gradual destruction of the previously formed “fibril chains” and the formation of ionic liquid clusters in the form of “streams” accumulating at the electrode (Figure 14i,j). After disconnecting the power supply, the tested mixture did not return to the initial state presented in Figure 14a—now the particles from the decay of “fibril chains” were permanently stuck at the electrodes (Figure 14j). In subsequent trials with the same portion of the mixture, no effects of the formation of any “fibril chains” were recorded. Multiple changes in the polarity of the power supply with an external during the electric field, also did not bring any effect. The tested PAO6 + CJ008 mixture underwent destruction and the same portion of did not show any ER effect, previously not in the rheological test (Section 3.3) and in the so-called dielectric spectroscopy (Section 3.4). However, the PAO6 + CJ008 mixture showed, in the range of the electric field strength *E* = 0~0.2 kV·mm^−1^, a much greater increase in viscosity compared to the second tested mixture i.e., GP1 + CJ001 (Section 3.3—Table 3 and Table 4).

The following photos (Figure 15a–h) show selected sequences showing changes in the internal structure of the mixture composed of the GP1 silicon damping liquid as a matrix and 2% (*v*/*v*) of the CJ001 ionic liquid, i.e., methyl-3-octyloxymethylimidazole tetrafluoroborate. This mixture was also tested only with an electric field strength *E* = 0~0.3 kV·mm^−1^, for the same reason as the PAO6 + CJ008 mixture. However, in this case, the process of creating and declining the ER effect was slightly different and much more dynamic. The obtained dispersion of the CJ001 ionic liquid in GP1 silicone oil was more uniform and homogenous (Figure 15a,b) than that of the CJ008 ionic liquid in the PAO6 base oil (Figure 14a,b), but the average particle size was larger. There was no such variation in mean particle size as before. As a result, the visible concentration of the particles of the CJ001 ionic liquid was slightly higher than that of the CJ008 ionic liquid. In the initial phase of the observation, when the electric field strength was equal to *E* = 0 kV·mm^−1^, the particles of the CJ001 ionic liquid did not change their position, size, or structure. Then, the observed sample was supplied with the electric field intensity *E* = 0.1 kV·mm^−1^. In the initial phase, single short chains appeared, in very short sections, arranged in the direction of the electric field, which joined together to form a stream of ionic liquid in the base oil, fulfilling the function of a “fibril chain” (Figure 15c,d). In the next phase, after increasing the electric field intensity to the value of *E* = 0.2 kV·mm^−1^, the formation of successive streams of ionic liquid took place (Figure 15e,f). After increasing the electric field strength to the value of *E* = 0.3 kV·mm^−1^, the chains were broken supply, the tested mixture did not return to the initial state presented in Figure 15a. Microparticles from the decay of the “fibril chains (streams)” adhered permanently to the electrodes, forming the layer visible in Figure 15g,h. Figure 15g shows the decay phase of the streams and the accumulation of the ionic liquid at the electrode. Figure 15h shows larger clusters of the ionic liquid lying at a short distance from the electrode. In subsequent tests with the same portion of the GP1 + CJ001 mixture, no ER effect was recorded, as in the PAO6 + CJ008 mixture. However, some differentiation of the course of the ER effect is visible, manifested mainly by different electrorheological characteristics (Section 3.3—Figure 4). As was said before, it was influenced not only by the different dielectric properties, but also by the physicochemical properties of the components of both mixtures. The recorded difference in the mechanism of the disappearance of the ER effect in both mixtures, especially the distinctly different course of the decay of the so-called “fibril chains (streams)” and the accumulation of the ionic liquid at the electrode, requires not only the previously discussed impact of the difference in viscosity of both components in the mixtures, but also the probable influence of their surface tension. It seems that the recorded effect is different for the GP1 + CJ001 mixture (Figure 15g,h) than for the PAO6 + CJ008 mixture (Figure 14i,j), as well as the differentiation of the electrorheological characteristics of the GP1 + CJ001 mixture in relation to the characteristics of the PAO6 + CJ008 mixture (Section 3.3—Figure 4), could also have been influenced by a significant difference in by a significant difference in the surface tension of the components of the GP1 + CJ001 mixture (Section 3.2—Table 1). It was found that at 25 °C the surface tension of GP1 oil was 19.547 mN·m^−1^, and the CJ001 ionic liquid was 26.032 mN·m^−1^, while in the second mixture the surface tension of both components was almost the same, for PAO6 was 27.523 mN·m^−1^ and for CJ008—28.099 mN·m^−1^. Additionally, the different rheological characteristics of both mixtures, i.e., GP1 + CJ001 (Bingham) and PAO6 + CJ008 (Newton) were directly influenced by the diverse chemical structure of their components. In the first case, one deals with a complicated structure of the GP1 damping oil, which is a mixture of polymethylsiloxanes with an admixture of CJ001, i.e., 1-methyl-3-octyloxymethylimidazolium tetrafluoroborate (Table 3), and in the second case, it was PAO6 hydrocarbon polyolefin base oil with an admixture of CJ008, i.e., trihexyltetradecylphosphonium bis(trifluoromethylsulfonyl)imide (Table 4). In both cases, it was not possible to obtain fully homogenous mixtures (Figure 14a and Figure 15a), which determined the occurrence of additional internal friction in the mixtures between their components. In the case of the PAO6 + CJ008 mixture, the formation of typical “fibril chains” was recorded (Figure 14) and much slower degradation as compared to the GP1 + CJ001 mixture was observed. GP1 + CJ001 mixture suddenly revealed the degrading CJ001 ionic liquid as exposed to the external electric field “streams” (Figure 15). From the chemical structure point of view, the components of the GP1 + CJ001 mixture are less compatible than the components of the PAO6 + CJ008 mixture. Moreover, the oxygen incorporated in the alkyl chain may cause its disintegration, which in turn may cause the mixture to lose its ER properties faster. The measurement technique may occur as an additional reason for the differentiation of the characteristic of both mixtures. In our study the rotational viscometer was used, the mixtures were subject to mechanical mixing, which may favor their disintegration. Previous own research [1,2,3,4,16] also showed that the tribological properties of the discussed ionic liquids are exceptionally good. The assessed lubricity parameters of these ionic liquids were much better, not only than base oils, but also brand engine and transmission oils. Anyway, information on very good tribological properties of various ionic liquids is quite common in the world tribological literature (e.g., [5,6,12,13,37,38]). Thus, the observed effect of ionic liquid accumulation at the electrodes, after the destruction of the “fibril chains” under the influence of an external electric field, may have a positive effect on the creation of the lubricating boundary film.

## 5. Conclusions

Summing up, on the basis of these extensive multifaceted studies of ER properties using the modernized Brookfield DV-III Ultra viscometer, studies in dielectric spectroscopy, and in situ microscopic studies of the mechanism of the ER effect, it can be concluded that:-it is possible to produce mixtures of hydrocarbon oils with a small admixture of an ionic liquid (here about 2% *v*/*v*), generating the ER effect under the influence of an external direct electric field, but due to the high conductivity of ionic liquids, their greater content in the mixtures leads to electric short-circuit;-in view of the great difficulties in obtaining a solution of an ionic liquid in hydrocarbon oil, it is very important to prepare a mixture of appropriate fragmentation and dispersion of the ionic liquid in the base oil in order to obtain a quasi-homogeneous liquid;-the basic mechanism responsible for the ER effect of mixtures of hydrocarbon oils with a small admixture of ionic liquids, exposed to an external direct electric field, is the formation of “fibrillary chains and/or streams” of micro/nanoparticles of the ionic liquid, “spread” between the electrodes along the line of the electric field;-the operating electric field after reaching a certain threshold value leads to the permanent destruction of “fibrillary chains and/or streams” and disappearance of the ER effect and the accumulation of ionic liquid particles at the electrodes (here at *E* = 0.3 kV·mm^−1^);-the course of the *τ* = f($\dot{\gamma }$) characteristic of a mixture of hydrocarbon oil with a small admixture of an ionic liquid, subjected to an external electric field, is primarily influenced by so-called dielectric properties of the electrically active component, or rather their change as a function to the applied BIAS (DC) voltage; smaller and slower change the permittivity of the ionic liquid allows to generate a much better ER effect;-it seems that the electrorheological characteristics *τ* = f($\dot{\gamma }$) the tested mixtures also depended on many physicochemical properties of their components and on the differentiation of their values including:>the difference in the viscosity of the insulating base oil and the added ionic liquid,>the difference in the value of the surface tension of the base oil and added ionic liquid;

In these studies, the Newtonian rheological characteristics were obtained for a smaller difference in the viscosity of the components of the PAO6 + CJ008 mixture, and the Bingham characteristic was obtained for a twice greater difference in viscosity of the components of the GP1 + CJ001 mixture; however, the maximum values of (*τ_max_*) and (*η_max_*) could have been additionally influenced by several times higher PAO6 base oil viscosity than GP1 oil, which could have contributed to the found slower degradation of “fibrillary chains and/or streams” of the CJ008 ionic liquid in PAO6 oil than CJ001 in GP1 oil.

The conducted research and analyzes have shown that the problem of using an ionic liquid as an ele4ctrically active component in an ER mixture is very complex and faces great difficulties not only because of their poor miscibility with most hydrocarbon hydraulic fluids and lubricating oils, but above all because of their large conductivity. However, it is not enough to try to use ionic liquids with very low conductivity. It is also necessary to analyze the course of their permittivity characteristics (*ε*) as a function of frequency (*f*) for different values of the voltage (*U*) of the BIAS (DC) field. The variability of the permittivity (*ε*) should be as low as possible. Therefore, there is still a need to produce and test fully homogeneous ER liquids, i.e., those in which the electrically active ingredient and the insulating base oil constitute a solution, then the mixing of both components is on a molecular scale, and then it is practically impossible to distinguish the components of such a liquid visually even under a microscope. The effects of an external electric field on such a homogeneous mixture are practically unknown. The literature information to date shows that only the Japanese managed to achieve a single success by producing and testing a homogeneous ER liquid composed of polymeric liquid crystals, but the increases in the viscosity of this liquid in an electric field were not impressive [37]. Among the ionic liquids, there are also those with a sufficiently long alkyl chain and to a certain temperature range, achieve a liquid crystal structure [38]. Another interesting research challenge is to undertake tests with such ionic liquids, with appropriately selected rheological and dielectric properties and enabling the production of a fully homogeneous mixture with the base oil. Due to the large number and variety of ionic liquids and their specific and unique properties, further extensive research on the possibility of their use, both in the electrorheological and tribological aspects, should be carried out. Meanwhile, in more recent publications on ER liquids, the authors still deal mainly with heterogenous liquids [39,40,41], in which the problem of sedimentation can be effectively minimized with an appropriate dispersing agent.

## Figures and Tables

**Figure 1 materials-16-00330-f001:**
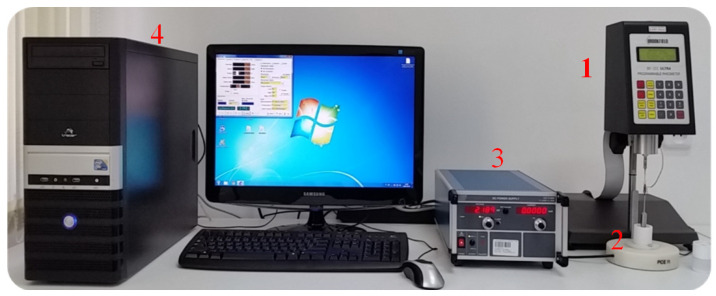
ER liquid testing stand: 1. Brookfield DV-III Ultra rheometer with insulated rotating spin-dle, 2. isolating adapter with cylinders equipped with electrodes, 3. high voltage power supply HCP 14-6500, 4. computer with Reocalc 32 v.2.6 control software.

**Figure 2 materials-16-00330-f002:**
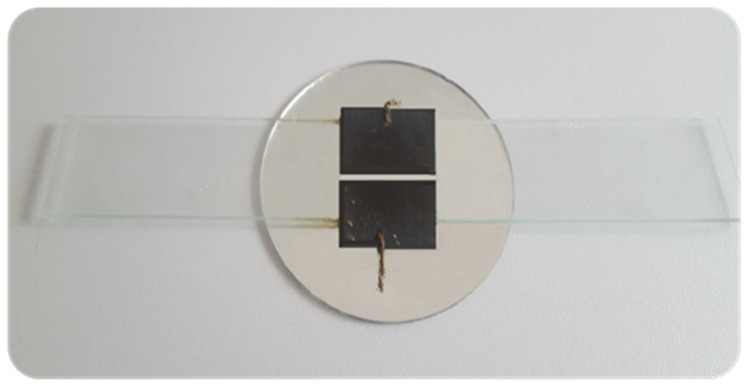
Glass vessel with electrodes for in situ observation of the ER effect mechanism.

**Figure 3 materials-16-00330-f003:**
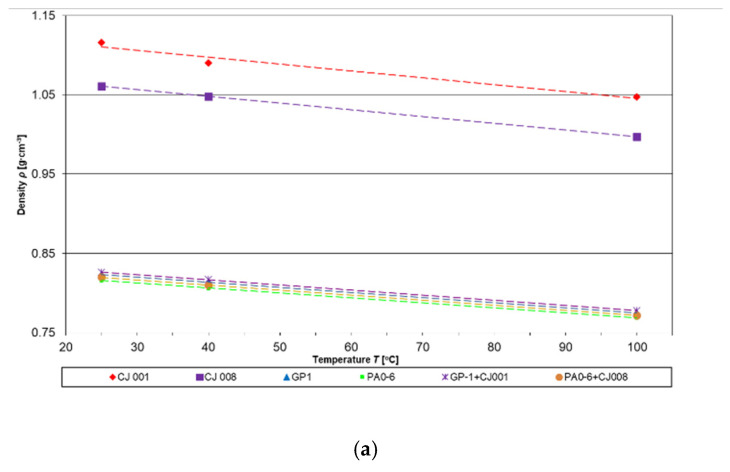
Dependence of the density (**a**) and viscosity (**b**) of base oils, ionic liquids, and their mixtures on temperature.

**Figure 4 materials-16-00330-f004:**
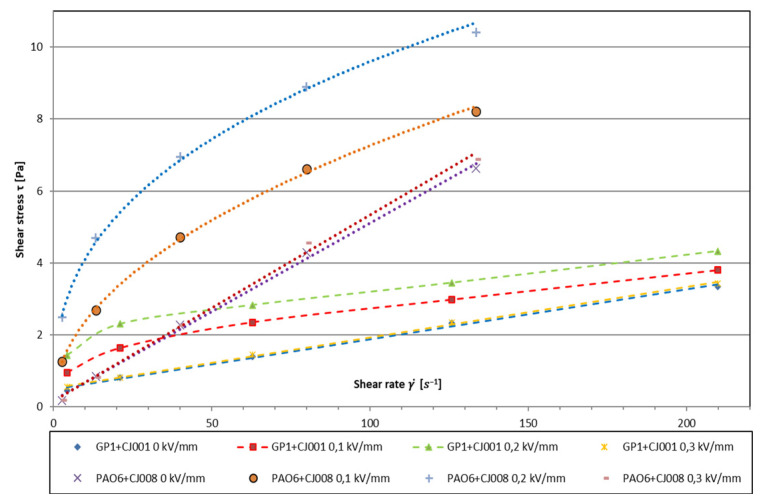
Characteristics *τ = f(*$\dot{\gamma }$) for mixtures GP1 + CJ001 and PAO6 + CJ008.

**Figure 5 materials-16-00330-f005:**
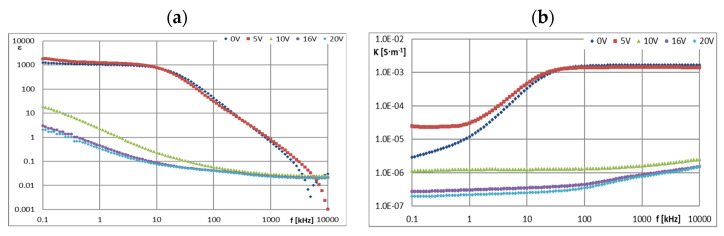
Characteristics: (**a**) *ε* = f(*f*) and (**b**) *κ* = f(*f*) for CJ001 at different values of *U*.

**Figure 6 materials-16-00330-f006:**
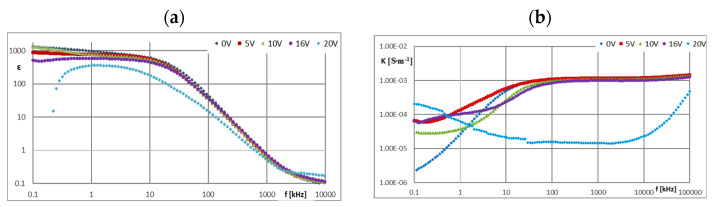
Characteristics: (**a**) *ε* = f(*f*) and (**b**) *κ* = f(*f*) for CJ008 at different values of *U*.

**Figure 7 materials-16-00330-f007:**
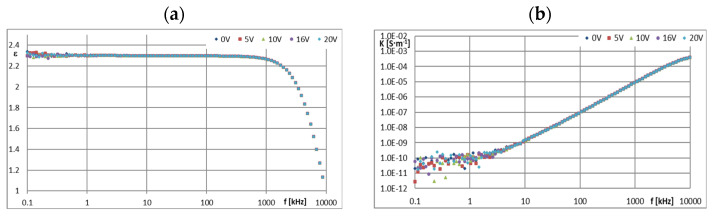
Characteristics: (**a**) *ε* = f(*f*) and (**b**) *κ* = f(*f*) for GP1 at different values of *U*.

**Figure 8 materials-16-00330-f008:**
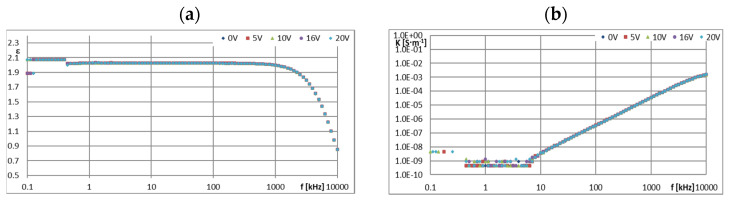
Characteristics: (**a**) *ε* = f(*f*) and (**b**) *κ* = f(*f*) for PAO6 at different values of *U*.

**Figure 9 materials-16-00330-f009:**
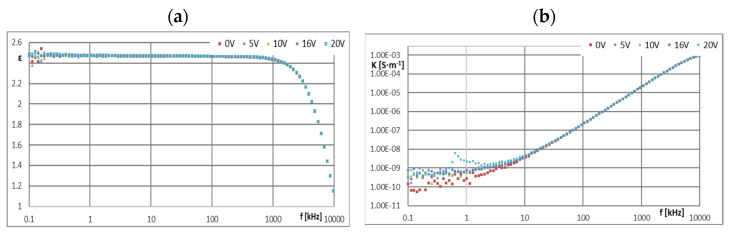
Characteristics: (**a**) *ε* = f(*f*) and (**b**) *κ* = f(*f*) for GP1 + CJ001 at different values of *U*.

**Figure 10 materials-16-00330-f010:**
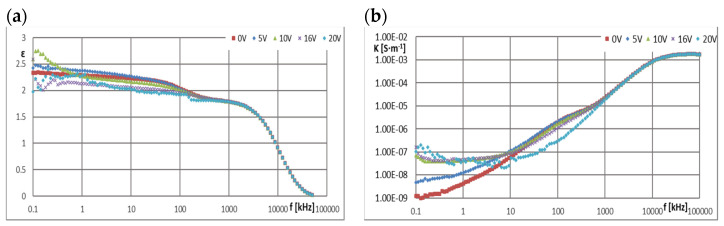
Characteristics: (**a**) *ε* = f(*f*) and (**b**) *κ* = f(*f*) for PAO6 + CJ008 at different values of *U*.

**Figure 11 materials-16-00330-f011:**
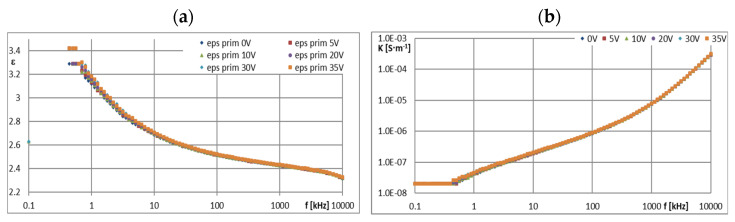
Characteristics: (**a**) *ε* = f(*f*) and (**b**) *κ* = f(*f*) for LID 3354S at different values of *U*.

**Figure 12 materials-16-00330-f012:**
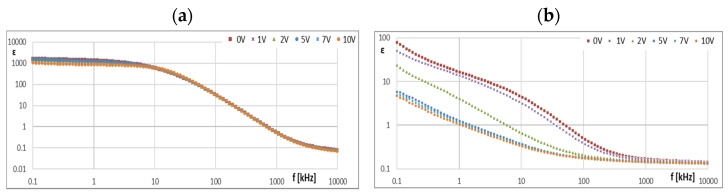
The results of the first measurement (**a**) and second (next) measurement (**b**) dependence of (*ε*) on the frequency (*f*) for the CJ008 sample in the BIAS field of low voltage (0~10 V).

**Figure 13 materials-16-00330-f013:**
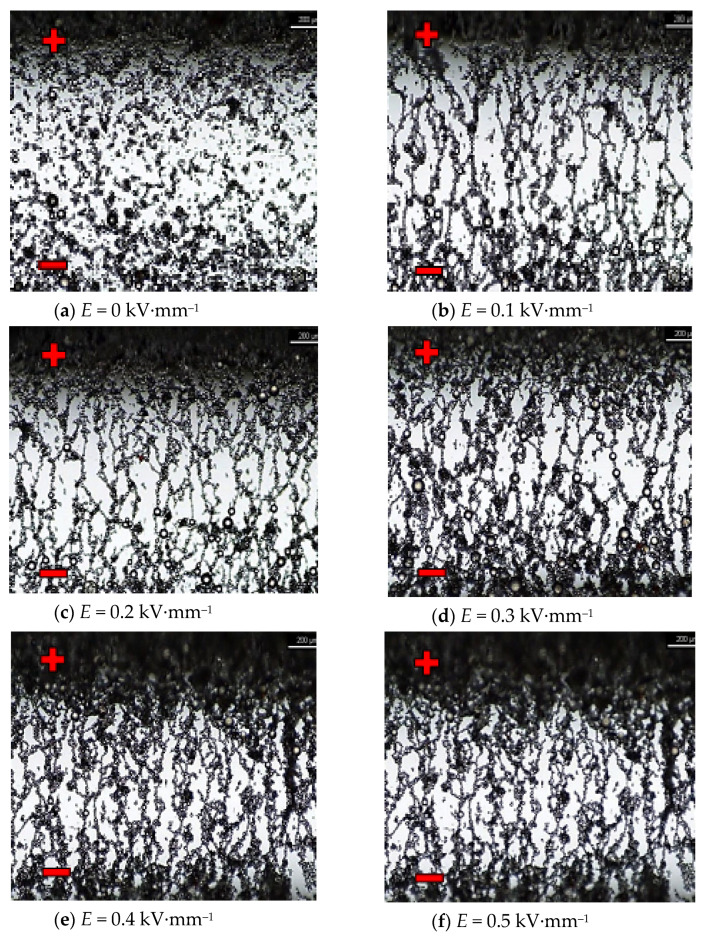
(**a**–**f**): Liquid LID 3354S in an electric field observed under a Nikon Eclipse LV100D polari-zing microscope.

**Figure 14 materials-16-00330-f014:**
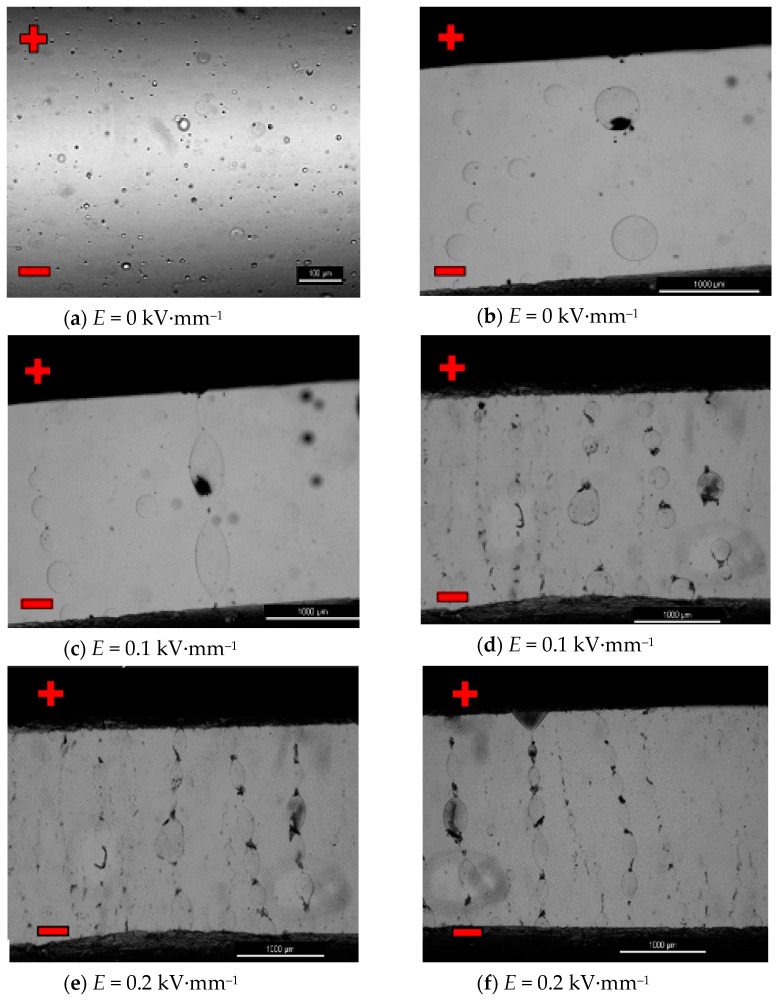
(**a**–**j**): Mixture PAO6 + CJ008 in an electric field observed under Nikon Eclipse LV100D pola-rizing microscope.

**Figure 15 materials-16-00330-f015:**
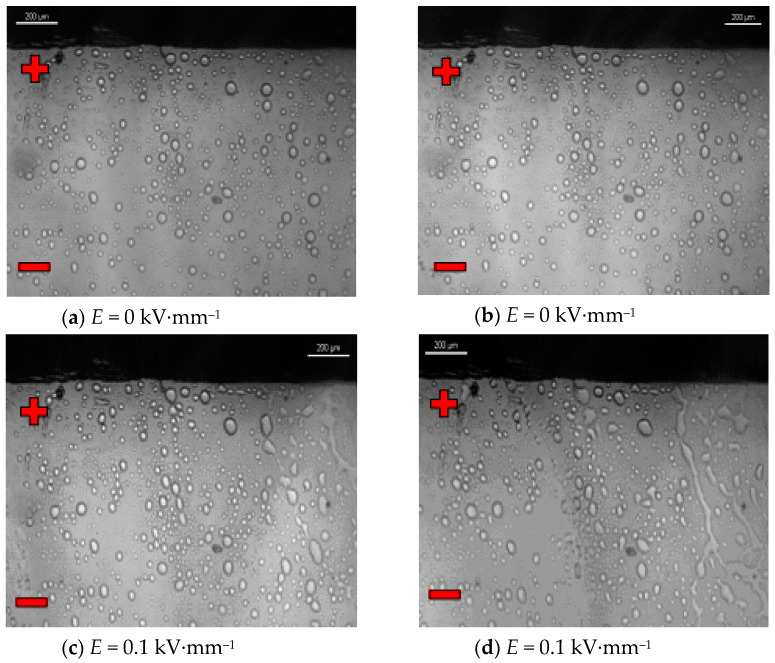
(**a**–**h**): Mixture GP1 + CJ001 in an electric field observed under Nikon Eclipse LV100D polari-zing microscope.

**Table 1 materials-16-00330-t001:** The basic properties of mixtures and their components determined at the temperature of 25 °C.

Liquid	ρ [g·cm^−3^]	υ [mm^2^·s^−1^]	η [mPa·s]	VI [-]	σ [mN·m^−1^]	n_D_ [-]	r_D_ [cm^3^·g^−1^]
CJ001	1.116	495.00	552.42	110	26.032	1.4304	0.2317
CJ008	1.061	308.43	327.24	127	28.099	1.4480	0.2523
GP1	0.823	15.33	12.61	264	19.547	1.4039	0.2971
PAO6	0.816	58.05	47.36	144	27.523	1.4558	0.3298
GP1 + CJ001	0.826	20.05	16.56	218	20.601	1.4061	0.2978
PAO6 + CJ008	0.820	61.08	50.08	140	27.998	1.4540	0.3302

**Table 2 materials-16-00330-t002:** Markings of sets “spindle/cylinder”.

Ordinal Number	Cylinder Diameter *D* [mm]	Spindle Diameter *d* [mm]	
		3.2	11.7
1	20.0	JJ1	JJ2
2	17.0	JJ3	JJ4
3	15.0	JJ5	JJ6
4	13.5	JJ7	JJ8

**Table 3 materials-16-00330-t003:** Results of research on ER properties of GP1 + CJ001 mixture.

Name of the Test Sample	Electric Field Strength E [kV·mm^−1^]	Spindle Speed n [rpm]	Shear Stress τ [Pa]	Shear Rate $\dot{\gamma }$ [s^−1^]	Dynamic Viscosity η [mPa·s]
Spindle/Cylinder Set					
GP1 + CJ001 JJ8 chemical structures: GP1 (polisiloxans): CJ001 (C_13_H_25_BF_4_N_2_O): 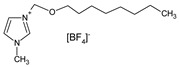	0 0.1 0.2 0.3	5	0.44 0.95 1.43 0.45	4.20	128.57 261.90 435.71 130.95
	0 0.1 0.2 0.3	25	0.81 1.63 2.30 0.81	20.98	39.08 77.69 109.63 38.61
	0 0.1 0.2 0.3	75	1.40 2.35 2.83 1.44	62.93	22.25 33.53 44.97 22.88
	0 0.1 0.2 0.3	150	2.33 2.98 3.45 2.35	125.85	18.51 23.68 27.41 18.67
	0 0.1 0.2 0.3	250	3.34 3.80 4.33 3.39	209.75	15.92 18.12 20.64 16.02

**Table 4 materials-16-00330-t004:** Results of research on ER properties of PAO6 + CJ008 mixture.

Name of the Test Sample	Electric Field Strength E [kV·mm^−1^]	Spindle Speed n [rpm]	Shear Stress τ [Pa]	Shear Rate $\dot{\gamma }$ [s^−1^]	Dynamic Viscosity η [mPa·s]
Spindle/Cylinder Set					
PAO6 + CJ008 JJ6 chemical structure PAO6 (polialphaolefins): 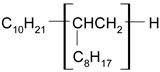 CJ008 (C_34_H_68_F_6_NO_4_S_2_P): 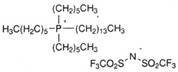	0 0.1 0.2 0.3	5	0.18 1.25 2.48 0.19	2.67	67.41 468.16 1247.19 71.16
	0 0.1 0.2 0.3	25	0.84 2.69 4.70 0.83	13.33	63.01 186.80 323.33 63.01
	0 0.1 0.2 0.3	75	2.27 4.72 6.94 2.28	39.98	56.78 105.55 131.06 57.03
	0 0.1 0.2 0.3	150	4.28 6.61 8.89 4.33	79.95	53.53 73.29 89.93 53.78
	0 0.1 0.2 0.3	250	6.63 8.21 10.41 6.69	133.50	49.66 60.75 74.31 49.89

## Data Availability

The data presented in this study are available on request from the corresponding authors.

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
