# Peer review of "Hydrocarbon Lubricating Oils with Admixture of Ionic Liquid as Electrorheological Medium"

_materials, 2022, doi:10.3390/ma16010330_

Round 1

Reviewer 1 Report

In this paper, authors characterized the electrorheological properties of the base oils with a small admixture of ionic liquids. However, scientific discussion should be added for publication.

1.     The viscosity of a liquid mixture is usually the sum of the product of the viscosity and volume fraction of each liquid. Why the dynamic viscosity of GP1+CJ001 was higher than that of PAO-6+CJ008 without applied voltage in table 3?

2.     Chemical structure of each component should be added in table3 or in other table. Why is one a Bingham liquid and the other a Newtonian liquid? Authors should add the discussion between the difference based on their chemical structure.

3.     Generally, The potential window of the ionic liquid is a few V. The ionic liquid supposed to be gradually breaking down during the

measurement.  Authors should make sure that ionic liquids are not decomposing during and after experiments under applied voltage. 

Author Response

Respectful Reviewer,

The authors want to express gratitude for the work of the Reviewer on improving being processed publication and its scientific overtones. We have tried to address all the valuable comments you have given us. We have made appropriate changes to the text of our manuscript.

We have included a detailed list of changes and responses to Your kind comments below:

  1. Reviewer comment:

In this paper, authors characterized the electrorheological properties of the base oils with a small admixture of ionic liquids. However, scientific discussion should be added for publication.

         Authors response:

The authors have enriched the discussion chapter and have added new current references (38-41).

  1. Reviewer comment

The viscosity of a liquid mixture is usually the sum of the product of the viscosity and volume fraction of each liquid. Why the dynamic viscosity of GP1+CJ001 was higher than that of PAO-6+CJ008 without applied voltage in table 3?

             Authors response:

In fact, this is usually the case with liquids that do not differ much in chemical structure and do not contain various additives. Otherwise, undesirable chemical reactions and uncontrolled changes in viscosity may occur. In this case, we are dealing with measurements on an upgraded PC with specialized Rheocalc32.v software. 2.6 (Brookfield Engineering Lubricants - CD PROG - 006 Contains Reocalc 32. v. 2.6 - Middelboro, MA, USA) enabling operation in the automated measurement mode, i.e. with automatic setting of selected angular speeds of spindle rotation. The modernized rheometer was calibrated (according to Reocalc 32.v. 2.6) for the assumed maximum rotational speed n=250 rpm, with the spindle fully immersed. The standard fluids were: Platinum 15W/40 engine oil, whose dynamic viscosity at 20oC is 199 mPa·s, and HL46 hydraulic oil, whose dynamic viscosity at 20oC is 120 mPa·s. As a result of measurements and appropriate calculations for the assumed rotational speed n=250 rpm, very good agreement was obtained [41]. Therefore, the viscosity for E = 0 kV mm-1, but for n=250 rpm (Table 3 and 4) should be compared with such an assessment. The listed viscosity value of GP1+CJ001 is lower than that of PAO6+CJ008.

  1. Reviewer comment:

 Chemical structure of each component should be added in table3 or in other table. Why is one a Bingham liquid and the other a Newtonian liquid? Authors should add the discussion between the difference based on their chemical structure. 

               Authors response:

According to the Reviewers comment the table 3 and 4 have been expanded. As suggested by the Reviewer, the structural formulas of the components of both mixtures are given in tables 3 and 4, in the first column (this column was modified, i.e. "chemical structure" was additionally entered in the header and the structural formulas of both components were entered below.

To explain the rationale for using Bingham and Newtonian liquid in our research the and pointing it out the to the Reader the authors include some text in Discussion chapter (lines   622-641).

  1. Reviewer comment:

Generally, The potential window of the ionic liquid is a few V. The ionic liquid supposed to be gradually breaking down during the measurement.  Authors should make sure that ionic liquids are not decomposing during and after experiments under applied voltage. 

            Authors response:

The purpose of these studies was to check the final effect, i.e. to find out whether it is possible to use ionic liquid as an electrically active ingredient in the ER mixture. It seems to be a complex problem and requires further research. The recorded degradation of "fibril chains/streams" and loss of ER properties suggest the possibility of ionic liquid disintegration. However, in this case, no detailed studies were conducted to confirm this suspicion. This problem is indirectly explained in the supplemented fragment of the text on the page 19 (lines 622 - 641).

We hope that you will find our answers and changes to the manuscript satisfactory and that the publication edited under your direction will be worth publishing in Materials

Kind regards,

authors

Reviewer 2 Report

Comments: materials-2021534

In the present investigations, the authors examine the experimental investigation of Hydrocarbon lubricating oils with admixture of ionic liquid as electrorheological medium. The topic of study is fascinating and well-developed. I advise the publication of the manuscript after some minor changes.

·         Please add some very recent papers related to your study.

·         Please add the novelty of the problem prior to the previous published work.

·         Abstract should be enhanced with some major results. If possible add the qualitative results.

Why authors consider CJ001 and CJ008? What is the advantage?

Author Response

Dear Reviewer,

The Authors thank You for Your thorough analysis of our manuscript. Your comments significantly increase the substantive overtones of our publication, which is why we tried to take them all into account. This is reflected in the changes made under your direction to the manuscript as well as in our responses below.

  1. Reviewer comment:

In the present investigations, the authors examine the experimental investigation of Hydrocarbon lubricating oils with admixture of ionic liquid as electrorheological medium. The topic of study is fascinating and well-developed. I advise the publication of the manuscript after some minor changes.

              Authors response:

The authors thank you for appreciating the importance of their research and its structure.

  1. Reviewer comment

 Please add some very recent papers related to your study.

              Authors response

The authors have included new references in the manuscript No 38, 39, 40, 41, reflecting (in their opinion) the latest trends in the discipline.

  1. Reviewer comment:

Please add the novelty of the problem prior to the previous published work.

               Authors response:

The authors included some new information to the manuscript to emphasize the novelty of the presented research (lines 15-17).

  1. Reviewer comment:

Abstract should be enhanced with some major results. If possible add the qualitative results.

              Authors response:

The abstract was enchanced with major results information as well as qualitative results (lines 32-37).

  1. Reviewer comment:

           Why authors consider CJ001 and CJ008? What is the advantage?

              Authors response:

The authors have had research experience with many ioniq liquids so far. Among them, several were purchased and used in earlier tribological studies (1-4, 7, 35). Based on the obtained results, the authors selected two ioniq liquids that enabled the preparation of quasihomogeneous mixtures.

We hope that you will find our answers and changes to the manuscript satisfactory and that the publication edited under your direction will be worth publishing in Materials.

Kind regards,

authors
